# Avoid Everything
# Model-Free Collision Avoidance with Expert-Guided Fine-Tuning

**Adam Fishman[1], Aaron Walsman[1], Mohak Bhardwaj[1], Wentao Yuan[1],**
**Balakumar Sundaralingam[2], Byron Boots[1], and Dieter Fox[1,2]**

[1]University of Washington
[2]NVIDIA Inc

**Abstract:** The world is full of clutter. In order to operate effectively in uncontrolled, real world spaces, robots must navigate safely by executing tasks around obstacles while in proximity to hazards. Creating safe movement for robotic manipulators remains a long-standing challenge in robotics, particularly in environments with partial observability. In partially observed settings, classical techniques often fail. Learned end-to-end motion policies can infer correct solutions in these settings, but are as-yet unable to produce reliably safe movement when close to obstacles. In this work, we introduce *Avoid Everything*, a novel end-to-end system for generating collision-free motion toward a target, even targets close to obstacles. *Avoid Everything* consists of two parts: 1) *Motion Policy Transformer* (M$\pi$Former), a transformer architecture for end-to-end joint space control from point clouds, trained on over $1,000,000$ expert trajectories and 2) a fine-tuning procedure we call *Refining on Optimized Policy Experts* (*ROPE*), which uses optimization to provide demonstrations of safe behavior in challenging states. With these techniques, we are able to successfully solve over $63\%$ of reaching problems that caused the previous state of the art method to fail, resulting in an overall success rate of over $91\%$ in challenging manipulation settings. Videos and our open source implementation are available at https://avoid-everything.github.io.

**Keywords:** Imitation Learning, Robotics, Collision Avoidance, Fine Tuning, Motion Planning

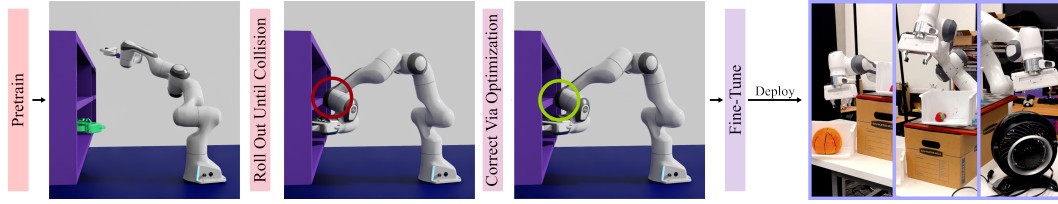

Figure 1: *Avoid Everything* is able to generate collision-free trajectories around complex obstacles in real time, using input from a single depth camera. It is pretrained on a large collection of expert demonstrations in simulated environments and fine-tuned with a technique we call *ROPE*, which actively seeks out collisions, corrects them via optimization, and uses the output as a training example for the network.

## 1 Introduction

The world is full of clutter. Humans effortlessly navigate through complex, unfamiliar spaces while constantly avoiding hazardous collisions. Robotics has not solved this key challenge, which is critical

8th Conference on Robot Learning (CoRL 2024), Munich, Germany.

to real-world success of robotic actors [1]. For robotic arms, this problem is especially pronounced due to their complex kinematics (see Fig. 1).

Many leading planning methods for collision avoidance rely on a stable, accurate, and fully observed representation of the robot's workspace. This allows planner [2, 3, 4, 5, 6, 7] invalidate or avoid invalid regions of the state space. However, building an accurate world model, particularly in cluttered spaces, is an open problem [8]. End-to-end imitation learning is an popular alternative technique that learns behavior without explicitly modeling the world, instead relying on patterns in how the expert behaves in response to the environment. However, these methods face the challenge of learning to avoid collisions from collision-free demonstrations alone. To address this, traditional motion planners can be used to track the paths produced by the network [9, 10] or directly incorporate a predefined world model into the learning framework [11]. These systems have the same limitations as traditional ones. When the world model is inaccurate, the system may collide. While expert demonstrations can be made collision-free, learning collision avoidant behavior necessitates a deep understanding of the interplay between the scene geometry and the robot's kinematics. Successful approaches have employed large datasets [12, 13] or explicit losses to encourage obstacle avoidance [13]. However, these techniques still fail in complex problems, leading to constrained capabilities [13] or continued reliance on traditional collision checking techniques [14, 11].

To address these challenges, we present *Avoid Everything*, an end-to-end [15] system that uses point clouds to generate goal-directed, collision-free motion for a robotic manipulator in cluttered 3D scenes. *Avoid Everything* uses a new network architecture *Motion Policy Transformer* (M$\pi$Former) that is trained end-to-end using expert supervision from a motion planner. Our model predicts single-step changes in joint configuration using point cloud observations, the robot's current configuration, and a target end effector pose. We also introduce a fine-tuning approach inspired by hard negative mining [16, 17]: *Refining on Optimized Policy Experts* (*ROPE*). *ROPE* is critical to reducing the collision rate in reaching toward the target. Through experiments, we show that *Avoid Everything* is able to safely solve over 63% of problems where the previous state of the art method [13] fails, resulting in an overall success rate of over 91% in challenging, partially observed manipulation settings. We also demonstrate that *ROPE* can be used as a general tool to reduce collisions, even in conjunction with DAgger [18], a standard technique for improving imitation performance.

Our contributions are as follows:

- *Motion Policy Transformer*, a new model architecture designed for predicting goal-directed robot motion from a point cloud and target location.
- *Refining on Optimized Policy Experts*, a novel fine-tuning algorithm for learning collision avoidance in robot motion generation.
- We demonstrate empirically that *Avoid Everything* reduces the collision rate of the previous state of the art by over 77% and improves success rate by 63%.
- We show that our method transfers well from simulated training to highly cluttered, real-world settings.

## 2 Related Work

**Reactive Control and Motion Planning**    Robot motion generation has traditionally been studied in the context of motion planning with a vast literature of methods [19, 20] based on graph search [4, 21, 22], sampling-based motion planning [2, 23, 24, 25, 26, 27], and trajectory optimization [28, 29, 30, 31]. See Appendix A for a more detailed discussion. While modern motion planning frameworks can achieve low control latency [32, 6, 33], they assume complete knowledge of the environment and make strong assumptions about obstacle representations for fast collision checking. Perception-driven reactive control of robots also has a rich history. Operational Space Control (OSC) methods such as [34, 35, 36] can enable robots to perform highly dynamic tasks at high control frequencies. However, their myopic nature can lead to local minima in the presence of obstacles. In a similar spirit to our work, Model-Predictive Control (MPC) approaches [37, 38] try to balance reactivity

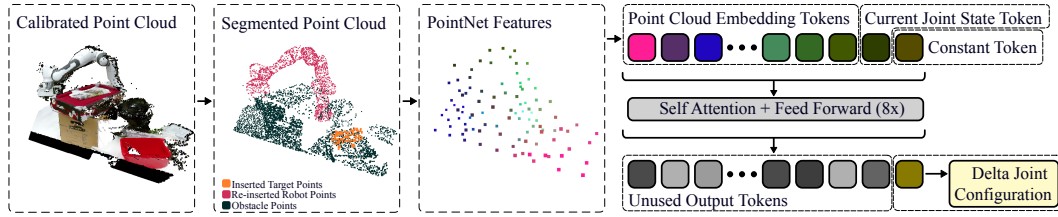

Figure 2: The input to MπFormer is a labeled point cloud, consisting of 4,096 points from the depth image (with robot removed), 2,048 points sampled from the robot mesh at the current configuration and 128 points from the gripper mesh placed at the desired target. The point cloud is encoded with 3 Set Aggregation [39] layers. The resulting features, along with an encoding of the current joint state and a learned query token, are passed through 8 transformer layers. Finally, the output token that corresponds to the query token is decoded into a delta joint configuration.

and planning horizon; however, real-time requirements often warrant the use of simple obstacle representations and short horizons that can still lead to local minima.

**Point Cloud Processing**    Point clouds, unordered sets of 3D points, are a lightweight and convenient 3D representation. Unlike other 3D representations such as meshes or Signed Distance Functions (SDFs), it is much easier and faster to obtain 3D point clouds from sensors such as depth cameras. As a result, many recent works choose to infer semantics and affordance from point clouds directly, skipping the need for 3D reconstruction. Leveraging powerful neural backbones that process point sets [39, 40], existing networks can segment objects [39], plan grasps [41] and check collisions [42] from partial point clouds only. Following the same spirit, in this work we show how to reliably generate collision-free joint trajectories from raw 3D point clouds.

**Imitation Learning**    Imitation learning describes a broad class of techniques to learn a policy from demonstrations, often made by a privileged expert [43]. Among imitation learning techniques, behavior cloning [44, 45] describes a set of techniques where a policy is directly trained to mimic an expert's actions. In manipulation, these actions are often phrased as end effector waypoints [9, 46, 10, 47], but these methods require a separate planner and collision checker to perform tasks safely. Recently [48, 49] have demonstrated strong capabilities for using transformers [50] to solve complex manipulation tasks with images as input and joint controls as output. Inspired by these methods, our architecture produces joint space controls given point cloud input.

Even when well-trained, learned policies typically exhibit small errors in prediction. As the error accumulates, the policy will encounter unseen regions of the state space, a problem often called *covariate shift*. Many techniques address this problem by strategically introducing a wider variety of states into the training dataset to increase coverage [51, 52, 18]. DAgger [18] augment the training data by providing expert demonstrations from states visited by a pretrained policy. Hard Negative Mining [16, 17] is a related technique in computer vision that augments the training data by labeling the explicit failures from a pretrained model. Our technique draws inspiration from both DAgger and Hard Negative Mining to explicitly correct the difficult states found from a pretrained model.

**Learned Motion Planning**    For the task of motion planning, imitation learning can be used either end-to-end or as a component of a traditional system. Some methods use learning to guide a traditional planner, either through a learned sampler [53, 54, 55, 56] or a learned heuristic function [57, 58]. Other techniques [38, 12] rely on a learned collision model [59]. Motion Planning Networks [14] uses a point cloud neural network to generate waypoints that are then verified with a traditional collision checker. Saha et al. [11] uses a diffusion model to produce plans based on the the SDF representation of the environment. Our neural architecture is most similar to Motion Policy Networks (MπNets) [13], which expects a segmented, calibrated point cloud and produces joint space controls. Despite its strong performance on a variety of problems, MπNets is trained with an expert that is smooth but incapable of reaching close to obstacles. As we discuss in Section 5.1.2, when the MπNets architecture is trained and evaluated on more challenging problems (using a more expressive expert), the policy often collides.

# 3 Methodology

In the following section, we describe our policy architecture, training implementation, and *ROPE*, our fine-tuning strategy that introduces hard negatives and explicit corrections.

## 3.1 Behavior Cloning for Collision Avoidance

*Avoid Everything* is a single-step policy that takes in a point cloud of the scene $P_t$, a 6-DoF target end effector pose $p$, and the robot's joint configuration $q_t$, where $t$ represents the current timestep. The scene point cloud $P_t$ consists of points sampled from the robot's arm at joint configuration $q_t$, points sampled from the surface of the obstacles, and points sampled from a mesh of the robot's end effector placed at the target pose $p$, as shown in Fig. 2. During training, the points are sampled entirely from the scene and robot meshes, and during inference, we use a depth image of the scene, mask out points associated with the robot, and insert artificial points for the robot and target sampled from the robot's meshes. This point cloud representation of joint state $q_t$ and target pose $p$ has superior performance over a numerical representation [13] due to the PointNet's [60] ability to understand point-wise relationships in local 3D space. The output of *Avoid Everything* at timestep $t$ is a delta joint configuration $\Delta q_t$, which is added to the current joint state $q_t$ to form a position target for the robot to follow.

### 3.1.1 Architecture

M$\pi$Former is a transformer architecture [50] that expects inputs consisting of a segmented, calibrated point cloud and the robot's current joint state. The point cloud consists of points representing the robot, the obstacles to avoid, and the robot's gripper placed at the target 6D pose. Each point is given a segmentation label representing which among these three types it belongs to. The points are first encoded through a PointNet++ [39], which compresses the initial point cloud into a sparse set of points with a wide feature vector. These sparse points are then flatted into a sequence to which we append an embedding of the robot's current joint state, as well as a constant token, the value of which is optimized during training. This sequence is then passed into an encoder-only transformer (see Figure 2). After 8 layers of self attention, we use the output token corresponding to the constant input and decode it through an MLP to produce the final output joint displacement, $\Delta q$. See Appendix B for a more thorough discussion of the architecture.

### 3.1.2 Loss Functions

We train *Avoid Everything* according to the same loss functions as M$\pi$Nets [13]: a task-space behavior cloning loss to encourage the policy to mimic the expert's behavior in task space, as well as a collision-avoidance loss. These losses are applied on predicted joint states, which are computed by adding the model's output (joint angle deltas) to the input joint angles and clamping the sum at the joint limits. See Appendix D for more detail on the losses.

## 3.2 Expert-Guided Fine-Tuning

After pretraining on a large dataset of expert state-action pairs, we observe that the policy is highly capable of reaching the target pose. Despite the reaching success, however, it still collides with objects in a significant percentage of problems in the held-out validation set (see Section 5.1.1). When we roll out the pretrained policy in simulation, we observe that the first obstacle penetrations are typically shallow and can be pulled out of collision by optimizing the configuration with respect to the collision loss (See Appendix Equation 2). Based on this observation, we introduce a novel technique of refining the pretrained policy for improved collision avoidance using fine tuning, inspired by Hard Negative Mining [17, 16] and trajectory optimization methods [28, 61, 31]. During the refining stage, we take mini-batches of random states from our training data—the same data used for pretraining—and roll out the pretrained policy for a fixed horizon. These trajectories can reach the target, collide, or neither. If the trajectory collides, we capture the state preceding the collision as input and optimize the colliding state to use as supervision. We then store this state-action pair in

Table 1: *Avoid Everything* vs. MπNets

| Planner | Cubby SR (%) / SCR (%) / RSR (%) | | Tabletop SR (%) / SCR (%) / RSR (%) | |
|---|---|---|---|---|
| *Avoid Everything* | 95.71 / 0.50 / 99.30 | | 91.97 / 1.03 / 98.44 | |
| MπFormer w. *ROPE* | 92.78 / 2.37 / 98.41 | | 89.57 / 4.15 / 96.82 | |
| MπFormer | 89.92 / 6.43 / 99.52 | | 86.00 / 11.26 / 99.57 | |
| MπNets w. *ROPE* | 87.35 / 4.72 / 96.68 | | 88.75 / 3.30 / 95.60 | |
| MπNets | 79.65 / 15.16 / 99.09 | | 77.95 / 14.72 / 95.69 | |

a buffer. If the trajectory does not collide, whether by successfully reaching the target or hitting the maximum rollout length, we use a separate buffer and store the input and output from the training batch, unmodified. In order to perform a weight update on the policy, we use a modified mini-batch made up of a fixed proportion $r$ of corrected examples and $1-r$ of unmodified expert examples. Once there are sufficiently many examples in both buffers, we remove these examples, assemble them into a modified mini-batch, and perform a weight update according to the losses used during pretraining. We then repeat this process with the updated policy. As discussed in Section J, increasing $r$ leads to lower collision rate but poor target convergence. We call this approach *Refining on Optimized Policy Experts (ROPE)*, and provide pseudocode in Algorithm 1 in Appendix E.

## 4 Data Generation Pipeline

We trained *Avoid Everything* on a large dataset of expert demonstrations in procedurally generated environments, examples of which are shown in Figure 3. The environments themselves were generated randomly and lie within two categories: 2x2 cubbies with randomized dimensions, cubby sizes, and world placement; and tabletops with a collection of randomly placed obstacles. All environments are constructed from primitives, which allows us to quickly sample point clouds during training. After generating each environment, we choose random problems in each en-

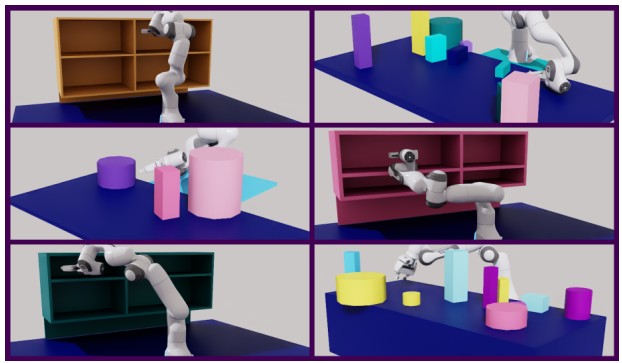

Figure 3: *Avoid Everything* is trained separately in two classes of procedurally generated environments, 2x2 cubbies. The first class is a 2x2 cubby with random dimensions and positions. The second is a table of varying dimensions a random set of objects placed on top. We generate expert demonstrations with AIT* [7] and spline-based shortcutting [62].

vironment and solve them with AIT* [7] with a 15 second timeout, followed by a spline-based shortcutting procedure [62]. For more detail on the planning pipeline, see Appendix C.

We trained *Avoid Everything* separately on each class of environments. For the cubby model, we used 1.25 million problems across 21,604 environments. For the tabletop model, we used 2 million problems across 43,646 environments.

## 5 Experiments

In order to evaluate *Avoid Everything*'s performance, we used a mix of quantitative experiments in simulation and qualitative tests on physical hardware. Our simulated experiments are in environments drawn from the same distribution as our training data. However, there are no shared environments between the evaluation and training problem sets.

### 5.1 Simulated Experiments

For the following experiments, we use two different evaluation settings. The first is a set of fully observed scenes where we sampled point clouds directly from the mesh. For these experiments, we

Table 2: Planner Performance in Partially Observed Scenes

| Planner | Perception | Cubby | | | Tabletop | | |
|---------|-----------|-------|---|---|----------|---|---|
| | | SR (%) / SCR (%) / RSR (%) | | | SR (%) / SCR (%) / RSR (%) | | |
| *Avoid Everything* | End-to-end | 92.34 / 3.10 / 98.68 | | | 87.62 / 4.28 / 97.70 | | |
| MπNets | End-to-end | 80.80 / 13.79 / 99.04 | | | 74.08 / 18.17 / 94.66 | | |
| RRTConnect | Octomap | 51.12 / 48.34 / 99.46 | | | 53.34 / 48.60 / 99.56 | | |
| AIT* | Octomap | 54.42 / 45.09 / 99.10 | | | 51.90 / 47.86 / 99.54 | | |
| CHOMP | Octomap | 49.38 / 36.67 / 77.98 | | | 75.72 / 19.22 / 93.74 | | |
| cuRobo | NvBlox | 73.06 / 22.88 / 94.74 | | | 76.86 / 22.11 / 98.68 | | |

used 10,000 problems for each environment type. As discussed in Section 4, our models are trained on fully-observed point clouds to allow for fast, on-the-fly data generation. We use these experiments to evaluate model features and fine tuning techniques.

We evaluate *Avoid Everything* against other planning techniques using partially observed point clouds generated with synthetic depth images. While our model was trained in fully-observed settings, our aim is for it to work robustly in partially observed environments. For these evaluations, we used 5,000 problems in each of our cubby and tabletop environments (10,000 total). For each environment, we captured a synthetic depth image from a randomly positioned camera facing the scene. While Fishman et al. [13] also evaluated Motion Policy Networks in partially observed settings, the viewpoint was fixed per class of environment. In order to evaluate the robustness to partial observability, we used random viewpoints for our synthetic images. See Appendix C for details.

**Metrics**  We show the results of these experiments in Tables 1, 2 and 3. Each table reports three metrics in each environment class. *Reaching Success Rate* (RSR) is the percentage of problems for which each method could provide a path (collision-free or not) to within $1$cm and $15°$ of the goal. *Scene Collision Rate* (SCR) is the percentage of these paths that had a collision with the scene, which we determined using discrete collision checking on the dense paths produced by each method. *Success Rate* (SR) is the percentage of problems that had a collision-free solution to the goal, including self-collisions. In addition to these top-line metrics, we also consider computation time, which determines the reactivity of each method. See Appendix G for a discussion of computation time.

### 5.1.1  Motion Policy Transformer

Without fine-tuning, MπFormer succeeds in $89.92\%$ and $86.00\%$ of our cubby and tabletop problems. However, after using DAgger and *ROPE*—the version we label *Avoid Everything* in Table 1—we see it succeed in $95.71\%$ and $91.97\%$ in the cubby and tabletop settings respectively. As discussed in Section 5.1.4, DAgger and *ROPE* improve performance independently, and the combination of the two leads to best-in-class performance.

In partially observed settings, we find that *Avoid Everything* still demonstrates strong performance, albeit with a slight performance drop. This robustness to perspective changes and incompleteness is a well-documented property of PointNet [60]. We expect that performance on partially-observable point clouds would improve if this data were included during training, but doing so at this scale would require significant additional computational resources.

### 5.1.2  Motion Policy Networks

Our system design is most similar to MπNets, which is the state of the art for learned end-to-end collision free motion. In order to evaluate our method, we trained MπNets on our expert data and compared it to MπFormer without any fine-tuning. We also fine tuned both models using *ROPE* and compared the performance. These results are shown in Table 1. Without any fine-tuning, we found MπFormer to outperform MπNets in both environments. Additionally, we find that *ROPE* significantly improves the performance of both models, reducing collision rates by more than half. However, after running *ROPE* on both algorithms, we find that the reaching success rate degrades more for MπNets through the fine-tuning process. MπFormer is better able to adapt to the hard

Table 3: MπFormer with Different Fine Tuning Strategies

| F.T. Stage 1 | F.T. Stage 2 | Cubby SR (%) / SCR (%) / RSR (%) | Tabletop SR (%) / SCR (%) / RSR (%) |
|---|---|---|---|
| None | | 89.92 / 6.43 / 99.52 | 86.00 / 11.26 / 99.57 |
| *ROPE* | | 92.78 / 2.37 / 98.41 | 89.57 / 4.15 / 96.82 |
| *DAgger* | | 93.19 / 4.08 / 99.54 | 89.17 / 5.59 / 99.31 |
| *DAgger* | *ROPE* | 94.63 / 1.10 / 99.59 | 91.10 / 2.45 / 98.41 |
| *Cons. DAgger* | | 94.88 / 1.28 / 99.16 | 91.06 / 2.31 / 98.74 |
| Cons. DAgger | *ROPE* | 95.71 / 0.50 / 99.30 | 91.97 / 1.03 / 98.44 |

negative examples without losing the ability to reach the target. We also compare the performance of *Avoid Everything* to MπNets in partially observed settings (Table 2) and observe that *Avoid Everything*'s collision rate is less than $\frac{1}{4}$ of MπNets's collision rate in these settings.

### 5.1.3 Classical Methods

While classical motion planners are highly capable of finding collision-free solutions, some even providing probabilistic guarantees [63], this hinges on the ability to verify states with a good perceptual model. Often, the scene is not fully observable, so these planners must rely on partial 3D reconstruction. While there are many ways to reconstruct a scene from a partial view, planning libraries still commonly recommend using analytic methods for 3D reconstruction. When using these analytic techniques, we found that planners often report a collision free path when, in fact, the path collides through an obstructed part of the scene. To understand this, we used four common planning implementations: RRTConnect [64], AIT* [7], and CHOMP [28] from MoveIt! [65] and trajectory optimization from cuRobo [6] along with their typical 3D reconstruction pipelines, OctoMap [66] for the MoveIt! planners and NvBlox [67] respectively. See Appendix H for details on our implementations. Note that these reconstructions used the partially observed point cloud—we attempted to use a state-of-the-art pretrained point cloud completion network [68] in conjunction with OctoMap, but found the completions to be inadequate for planning (see Appendix I). Our partially observed evaluations use randomly sampled camera viewpoints and in these settings, we found that all four planners tend to collide, despite the fact that they are highly effective at finding paths they believe are feasible. Notably, RRTConnect and AIT* collide most frequently (over $48\%$ and $45\%$, respectively), likely due the random sampling, whereas the trajectory optimization methods collide less—CHOMP collides in over $19\%$ of scenes and cuRobo collides in over $22\%$—likely due to the optimization objectives encouraging to stay away from visible obstacles. Meanwhile, for the same problems, *Avoid Everything* collides in less than $5\%$ of problems.

### 5.1.4 Fine Tuning Performance

After pretraining MπFormer, we evaluated several techniques for fine tuning the model. The results of these experiments are shown in Table 3. See Appendix for more detail on each method.

***ROPE*** When using *ROPE*, we found that it's important to balance corrected states (henceforth referred to as *ROPE* examples) with expert demonstrations, *i.e.* from pretraining, within the batches. When fine-tuned on *ROPE* examples alone, we found collision rates dropped to zero, but reaching error increases to over $8$cm. The results shown in Table 3 use a ratio of $20\%$ *ROPE* examples in each update batch, which dropped collision rates by approximately half without significantly increasing reaching error. However, for downstream tasks where precise reaching is less important, the *ROPE* ratio could be increased to improve safety. See Appendix J for more discussion.

***DAgger*** DAgger [18] is a highly effective technique to address covariate shift in imitation learning. After pretraining the policy and rolling it out, DAgger queries the expert for instructions from each achieved state. Although DAgger is useful to improve policy performance, it is can be computationally infeasible with an expensive expert. Whereas DAgger queries for a demonstration at every state, *ROPE* only corrects the difficult states. And, instead of using the original expert, *ROPE* relies only on local optimization, which is comparatively fast. We evaluated two versions of DAgger–one that

uses the same, pretraining expert and one that uses a more conservative expert, *i.e.* one that always stays at least 2cm from obstacles, which we label Cons. DAgger. The conservative expert cannot solve every problem and we simply remove any unsolvable problems from the fine-tuning dataset. Despite this, we find that the conservative version of DAgger is a more effective fine-tuning strategy. We attribute this behavior to the fact that the pretrained network has always learned the necessary kinematic behavior to reach targets in close proximity and the conservative expert then demonstrates that, outside of these cases, the policy should veer away from obstacles. After using either version of DAgger, we find that using *ROPE* further reduces collisions and improves success, demonstrating that these two methods correct for different types of failures. We discuss these findings, along with our DAgger implementation, in greater detail in Appendix J.

### 5.2   Performance on Real Robot Hardware

We deployed *Avoid Everything* on a Franka Emika Panda robot using point clouds from a calibrated depth camera. Key results are summarized here, with additional details in Appendix K.

We observed that the model is excellent at avoiding obstacles on the table when those obstacles are at least partially observable by the camera. We commonly saw collisions into obstacles that were fully occluded or out of the camera's field of view. We expect this issue could be improved with additional cameras to obtain a more complete point cloud. Many of the obstacles placed in front of the robot were far outside the training distribution, yet the model was able to avoid them easily. However, we found that highly complex obstacles, particularly those with thin structures (e.g. an office chair on its side) can result in collision. Not only was this obstacle out of distribution, but the rear legs were unobserved by the camera, leading to a compounding of our two main challenges.

We also observed signs that the model generalizes outside of its training data to produce safe behavior within highly novel settings. When we placed the target inside an obstacle, the model tends to hover above the obstacle without attempting to go in. This is despite the fact that none of the targets in training were ever inside obstacles. However, while this behavior occurred in the majority of cases, the robot did sometimes try to push through an obstacle to reach a target. We noticed this most often when the top face of the obstacle that unobservable by the depth camera, leading the model to think the object was an open bin instead of a closed box.

## 6   Limitations

Avoid Everything can achieve low collision rates in complex environments, but challenges remain. A key issue is generalization—while it performs well on in-distribution tasks, it struggles with unseen obstacle configurations and target poses outside the training distribution. The simple, gradient-based optimization used for fine-tuning may also be insufficient in highly complex settings, requiring more advanced techniques. Like other black-box systems, Avoid Everything lacks guarantees; in fully observed scenarios, future work could combine Avoid Everything with traditional planners to ensure safety. Finally, training requires substantial data and compute, which is costly and environmentally harmful.

## 7   Conclusion

*Avoid Everything* is an end-to-end system that can create safe, collision-free motion toward a goal using only a partially observed point cloud. The system consists of two novel components, MπFormer and *ROPE*. MπFormer is an end-to-end transformer architecture that produces joint space controls toward a target. With no fine-tuning, MπFormer is more capable than the existing state of the art for end-to-end motion generation. *ROPE* is a fine-tuning technique that leverages optimization to correct states where the pretrained policy collides. When used for fine-tuning, *ROPE* improves policy performance of both MπNets and MπFormer. *Avoid Everything*, the combination of MπFormer and *ROPE*, far outperforms other methods at generating end-to-end collision-free motion. See our website https://avoid-everything.github.io for videos of our policy and our open source implementation.

## Acknowledgments

This research was made possible by the generosity, help, and support of many. Among them, we would like to thank the following people for their contributions. Thank you Chris Xie for sharing your knowledge of transformers generally, as well as how to apply them effectively for 3D vision. Thank you Mitchell Wortsman for your help in tuning our optimizer to have useful loss curves. Thank you Jiafei Duan and Neel Jawale for your help with the real robot infrastructure and experiments. Thank you Daniel Gordon for sharing your wealth of deep learning expertise. Thank you Zak Kingston for your help improving and optimizing our expert pipeline. Thank you Rosario Scalise and Matthew Schmittle for helping to brainstorm our planner design. Thank you Adithyavairavan Murali for sharing your expertise on hard negative mining. Finally, thank you Jennifer Mayer for your help in organizing the paper into a cohesive story.

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

# Appendix

## A    Related Work

**Analytic Motion Planning**    Complex, high degree of freedom robots, such as manipulators, are often not well suited to a predefined graph, and instead benefit from continuously sampling the space to form a graph and a corresponding path through it [64, 23, 7]. Trajectory optimization [32, 69, 70, 28] is another approach well-suited to manipulation, that instead uses cost functions to refine an initial path. This often leads to trajectories that are qualitatively preferable to those of sample-based planners, but these algorithms are often subject to local minima. Whereas sampling-based planners reject states in collision, trajectory optimization ensures safety through constraints. Some techniques leverage hard constraints [69, 32, 71], while others approximate hard constraints with highly-weighted penalties on undesirable behavior [6, 28, 72].

**Safety**    Given the inherent risks of robot operation, safety has been an important component of robot planning for many years. Classical approaches to safety, such as reachability analysis [73] can provide powerful guarantees. While these methods have been successfully used in applications such as aircraft landing [74], refueling [75] and vehicle planning [76], they are challenging to apply in robotic settings with a large state space for computational reasons.

Given this difficulty, many researchers have turned to learning systems which generate signed distance functions [77], radiance fields [78] or implicit surfaces [79] which can be used as safety representations for motion planning and control [80, 81, 82, 83]. Our approach skips the potentially costly reconstruction of these representations and instead learns to predict actions directly from point cloud observations in an end-to-end fashion.

## B    Architecture

M$\pi$Former uses PointNet++ [39] to encode the point cloud and a transformer [50] to fuse the point cloud features with a representation of the current joint state. The input point cloud has a feature vector of length 4 for every point. All obstacles are assigned the same feature, all target points are assigned the same feature, and each robot point, which are sampled deterministically from the robot's mesh, is assigned a unique feature to disambiguate points on the arm. Our PointNet++ encoding architecture consists of three Set Aggregation (SA) layers. SA layers are a sparse 3D analog to convolutional layers. Each layer receives a point cloud where each point has a feature and outputs a smaller point cloud by using furthest point sampling to select $\frac{1}{4}$ of the points. Then, each sampled point is used as the center of a ball query. The ball query samples up to 64 points inside the ball and concatenates the ball center's coordinates to each point's feature vector. A four-layer MLP is then run on each point and MaxPool [84] collects the points inside the ball to produce a single feature per ball. The layers' ball queries have radii of 5, 30, and 50 centimeters respectively. Our input point cloud always has 6,272 points–4,096 obstacle points, 2,048 robot points, 128 target points. The downsampled point cloud after the third set aggregation layer has 98 points. Finally, we add 3D positional encoding to each of these 98 points, similar to [41].

The transformer takes a sequence of tokens as input, consisting of the 98 output features of the third SA layer, a token for the current joint configuration, and a learned constant token, similar to the decoder tokens in [48]. We get the joint angle token by passing the joint angles, which are normalized to be between -1 and 1, through a single linear layer. Our transformer has 8 layers with an embedding dimension of 512 and a feed-forward dimension of 2,048. To produce the final output $\Delta q$, we take the last token of the output sequence and map it through a single linear layer.

## C    Data

Out environments are similar to those demonstrated in M$\pi$Nets, but they differ in two key ways: we augmented the cubby design to encourage reasonable expert behavior by adding a floor beneath

the robot, and we increased the complexity of the tabletop environment by adding more objects and increasing the range of reachable poses. Within these constructed environments, we randomly sample end effector poses and their corresponding inverse kinematics (IK) solutions, which we compute using IKFast [85]. For the cubby environments, the poses are all grasping positions inside a cubby. For the tabletop, the poses are grasps pointing toward the lower hemisphere and placed either near the table's surface or on top of the objects. We also add neutral configurations drawn from uniform distribution around the robot's default pose to the tabletop data. These poses, for both types of environments, must be at least 5mm away from obstacles. We then use AIT* [7] with a path-length objective combined with a spline-based shortcutting [62] to generate expert demonstrations. In our planning pipeline, we impose a 20 second time limit in which we sample uniformly from the robot's configuration space, marking any sample that is either in self-collision or within 5mm of an obstacle as invalid. During the smoothing stage, we fit a collision and dynamics-aware spline to the planned path while shortcutting. We then sample from the spline at a fixed timestep, leading to paths with similar velocities, but varying lengths.

We chose this sampling-based pipeline because it enables us to produce expert demonstrations that lie precariously close to obstacles. Previously, M$\pi$Nets [13] demonstrated strong performance when trained with a so-called *Hybrid Expert*, which uses a reactive controller [86] to follow a planned end effector path. While this expert is effective for learning, it is highly conservative, preferring to stay far away from obstacles. In their experiments, the authors demonstrated that the *hybrid expert* demonstrations are insufficient to learn to solve problems that lie very close to obstacles. With our sampling expert, we chose a 5mm buffer from obstacles because this is sufficiently close for most tasks. As we designed our expert, we observed that increasing the collision margin improves learned collision avoidance, but this limits the expert's ability (and thus, the policy's ability) to plan to targets near obstacles.

When generating partially observed point clouds during inference, we captured depth information from randomized camera positions placed in the scene. In these scenes, we placed the robot at a fixed neutral starting configuration and segmented the robot out of the image. To randomize the camera, it was first placed in the scene at a predefined location facing the robot and obstacles, and was then rotated randomly by up to 30° about the z-axis (rotating side to side), then again by up to 10° about the camera's local x-axis (tilting up and down). Both of these rotations were applied using a fixed pivot point directly in front of the camera. Finally, the camera was translated randomly along the global z axis and y axes by up to 25cm.

To generate our expert dataset, we used a single desktop with a AMD Ryzen Threadripper 3990X 64-Core Processor. Generating the cubby and tabletop data took four and six days respectively.

## D    Loss Functions

**Task Space Loss**    The aim of this loss is to compare the physical positions of the policy's predicted robot joint space configuration and the expert's joint space configuration. For both configurations, we use forward kinematic functions $\phi^{\{i\}}(\cdot)$ to map joint angles of the robot $q$ to 1,024 points $x^{\{i\}}$ on the robot's surface, represented in 3D coordinates.

$$L_{\text{BC}}(\hat{\Delta}q) = \sum_{i=0}^{1,024} \|\hat{x}^i - x^i\|_2 + \|\hat{x}^i - x^i\|_1 \tag{1}$$

Like M$\pi$Nets, we sum $L1$ and $L2$ distances in the loss because the sum penalizes both large and small errors. We use a task space loss following M$\pi$Nets, which demonstrated it to be more effective when reasoning about collision avoidance as small perturbations along the kinematic chain can lead to large deviations for the end effector.

**Collision Avoidance Loss**    The training data was generated in simulation, giving us access to privileged information unavailable during inference, including a signed-distance representation of the scene. To avoid collisions, we use a hinge-based loss on $D(x)$, the signed distance from a point

$x$ on the robot to the nearest surface in the scene. Inspired by motion optimization [28, 31, 87], this loss effectively pushes the robot out of regions of collision. As in Equation 1, we use 1,024 points $x^{\{i\}}$ on the robot's surface to measure collision.

$$L_{\text{collision}} = \sum_i h(\hat{x}^i), \text{ where}$$

$$h(\hat{x}^i) = \begin{cases} -D(\hat{x}^i), & \text{if } D(\hat{x}^i) \leq 0 \\ 0, & \text{if } D(\hat{x}^i) > 0 \end{cases} \tag{2}$$

## E    ROPE

Our expert-guided fine tuning algorithm *Refining on Optimized Policy Experts (ROPE)* refines a pretrained model to reduce the collision rate using automated labeling of data generated by the learning agent. This algorithm first rolls out a fixed length horizon sequence $s'$ using the current model. To start a rollout, we randomly sample an expert trajectory from our training data and a state along that trajectory. We then roll out the expert for up to 50 steps. If this rollout collides at any point, we terminate the rollout and generate a fine tuning example by using the state preceding the collision as input and optimizing the colliding state to use as output. This optimization uses the collision avoidance loss in Equation 2 to push the state out of collision. We use AdamW to perform this optimization for simplicity, although we expect other methods typical to motion optimization such as Gauss-Newton or Levenberg-Marquardt may lead to a faster fine-tuning procedure. We continue to roll out sequences until enough corrected data has been collected to form a batch, after which the model is fine tuned using the task space and collision avoidance losses outlined in Appendix D. We then repeat this process with the newly updated policy. Algorithm 1 provides pseudocode. During fine-tuning, we continually use the latest policy to perform rollouts, even as it is updated. In our best-performing fine-tuning experiment, we reached peak performance after 21 hours of training.

## F    Training Implementation

*Avoid Everything* was trained on an NVIDIA 4090 in batches of 50 using AdamW [61] with a learning rate of $5e-5$ and a linear warmup of 5000 steps from $1e-5$. On the cubby environment, the model was trained for 1.2 million steps, which took approximately four days.

During training, we add small amounts of random noise to the input configurations, which [52] showed leads to improved robustness. Like MπNets, the training scenes are constructed from primitives, so point clouds can be generated on the fly during training by sampling points from the surfaces of these primitives. Robot points are sampled deterministically from the mesh of the robot. When *Avoid Everything* runs on the real robot, we mask out the robot points in the depth cloud and re-insert them using the same deterministically sampled points from training.

## G    Computation Time and Reactivity

Despite its high success rate, *Avoid Everything* is less suitable than MπNets for high-frequency control. Running on a NVIDIA 4090 GPU, we can run *Avoid Everything* at 33Hz, meanwhile MπNets runs at 150Hz. While this discrepancy could be improved with a more optimized transformer implementation, the *Avoid Everything* architecture requires a relatively expensive pass through self-attention, whereas MπNets uses average pooling to aggregate PointNet features from the point cloud encoder, which is much less computationally expensive.

When compared to traditional planning pipelines under partial observation, *Avoid Everything* shows both significantly improved collision rates as well as much higher reactivity. For *Avoid Everything*, the computation cost for every action is the same, whether or not the scene changes. Meanwhile, traditional planners have to recompute a path when the world changes. When the scene is static, traditional pipelines only need to run once because they produce a full path with each call. Even in these cases, we found *Avoid Everything* to produce a full path faster than the MoveIt! planners.

**Algorithm 1:** Refining on Optimized Policy Experts

**Result:** $\pi$

1   $\pi \leftarrow \pi_{\text{pretrained}}$
2   $b \leftarrow$ Batch Size
3   $r \leftarrow$ Correction Ratio
4   $D_{expert}$ ▷ Dataset containing expert demos
5   $B_{coll} \leftarrow \{\}$ ▷ Collision correction demos
6   $B_{free} \leftarrow \{\}$ ▷ Collision-free expert demos
7   **for** {state, next_state, tgt, scene} **in** $D_{expert}$ **do**
8     $s \leftarrow$ state
9     **for** $j \leftarrow 1$ **to** $N$ **do**
10       $s' \leftarrow \pi(s, \text{tgt})$
       ▷ If $s'$ collides, correct & add to buffer
11       **if** COLLIDES($s'$, scene) **then**
12         $\bar{s}' \leftarrow$ CORRECT($s'$, scene)    ▷ Apx Eqn 2
13         ADD($B_{coll}, \{s, \bar{s}', \text{tgt}, \text{scene}\}$)
14         **break**
15       **end**
       ▷ If rollout finishes without collision, add original example to buffer
16       **if** REACHED($s'$, tgt) **or** $j = N$ **then**
17         ADD($B_{free}, \{\text{state}, \text{next\_state}, \text{tgt}, \text{scene}\}$)
18         **break**
19       **end**
20       $s \leftarrow s'$
21     **end**
22     **if** $|B_{coll}| > rb$ **and** $|B_{free}| > (1-r)b$ **then**
      ▷ Make batch & clear buffers
23       $B \leftarrow \{\text{POP}(B_{coll}, rb), \text{POP}(B_{free}, (1-r)b)\}$ ▷ Compute loss, gradient update
24       $\pi \leftarrow$ UPDATE($\pi, B$)
25     **end**
     ▷ Reached validation accuracy or timeout
26     **if** TERMINATION_CONDITION($\pi$) **then**
27       **terminate**
28     **end**
29 **end**

Curobo, however, was the fastest option in these scenarios, and was able to produce an entire path nearly as quickly as *Avoid Everything* produced a single action (See Figure 4).

## H   Partial Observability for Analytic Planners

Figure 5 show examples of the perceptual pipelines we used for MoveIt! [65] planners (RRTConnect [64], AIT*[7], and CHOMP [28]) and cuRobo [6]. MoveIt! is a popular motion planning library that integrates natively with Octomap [66] for perception. We used an Octomap with a resolution of 5mm. Our implementation of RRTConnect [64] used with a 5s timeout. When using CHOMP [**?**], we use the same RRTConnect parameters to find the seed trajectory. For AIT* [7], we specify a time out of 20 seconds and a configuration space path length objective, but we terminate the search as soon as a feasible trajectory. Note that this is different than the AIT* implementation we use for generating training data, where we use a full scene model and continue to refine the trajectory for 20 seconds.

In the cubby settings, we found that RRTConnect found a solution in $99.46\%$ of the problems and we attribute the remaining to noise that could be addressed with a longer timeout. However, of these successful plans, over $48\%$ of them had collisions. We found that AIT* had slightly lower collision rates, $45.09\%$ and $47.86\%$ in the cubby and tabletop settings respectively, which we expect has to do with the adaptive heuristic used for search. Overall, the performance of both methods was very similar, but despite their similar performance in these baselines, we expect that AIT* could be tuned

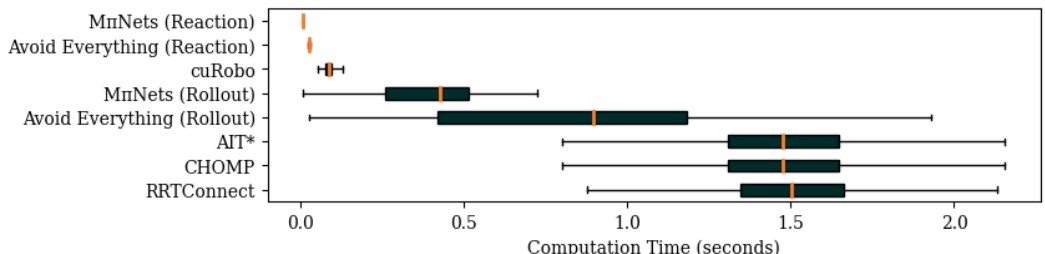

Figure 4: This figure depicts the end-to-end computation time needed to run *Avoid Everything*, along with the baseline methods, to solve partially observed problems in the cubby setting. Both *Avoid Everything* and MπNets [13] provide low-latency inference in dynamic settings, whereas the model-based methods each require an expensive call to a perception pipeline that increases latency. While MπNets is fastest, *Avoid Everything* is much faster than our other baseline methods when used for reactive control. In static scenes where an open-loop rollout is appropriate, cuRobo (together with NvBlox) is the fastest option.

to outperform RRTConnect through improved cost function design and by giving it more time to optimize its objective.

We ran similar tests using two different trajectory optimization methods designed to produce smooth trajectories, CHOMP [?] combined with Octomap [66] as well as cuRobo [6] combined with NvBlox [67]. CHOMP finds a path in 77.98% of cubby trajectories, but 36.67 of these trajectories have collisions. CHOMP performs better in the tabletop setting, likely because it has fewer bug traps that are often challenging for local optimization techniques. Meanwhile, cuRobo finds a path in 94.74% of of cubby problems, but 22.88% of these trajectories have collisions. We set the nvBlox resolution to 1cm for this test after consulting with the authors of cuRobo [6].

An advantage of classical methods such as those in our baselines is that they did not require special tuning or training for either environment. While we expect that their performance could be improved with additional tuning, the default parameters exhibit similar performance in both settings. Despite *Avoid Everything* having stronger performance in both environments, we do not expect it to generalize to wholly new settings as classical methods can.

## I   Point Cloud Completion with Classical Pipeline

When capturing point clouds with a depth camera, obstructions in the scene create holes in the point cloud. As discussed in section 5.1.3, classical methods often produce a valid path through the observed point cloud but collide with the scene in the unobserved regions. This problem is particularly pronounced in our RRTConnect [64] baseline because the planner searches for any valid feasible path by sampling in free space. Since the unobserved regions are registered as free space, the planner is just as likely to plan through these regions as any other free space in the scene. Instead of using OctoMap to directly represent the points captured from the camera, we could instead use a point cloud completion network, such as the state-of-the-art method AdaPoinTr [68], to estimate the completed shape of the point cloud before constructing the OctoMap and using it for planning. However these techniques are subject to their training distribution and are typically trained on specialized datasets such as ShapeNet [88] and do not generalize. We attempted to use this strategy as a baseline, but found that when pretrained with the Projected ShapeNet-55 dataset, the AdaPoinTr model cannot accurately complete our scenes (see Figure 6), leading to low success rates for the planner. This was particularly pronounced in the cubby setting, where the RRTConnect planner's reaching success rate (RSR) was 8.84% and among these solutions, the scene collision rate (SCR) was 80.09%. This is a significant degradation from using OctoMap without completion where RSR is 99.52% and SCR is 67.16%. The low planning success rate after completion is largely due to the fact that the completed point clouds obscured either the starting configuration or target pose, making it impossible to find a valid plan. Point cloud completion performed better in the tabletop settings, where the RSR is 74.14% and SCR is 41.03%. However, these metrics are

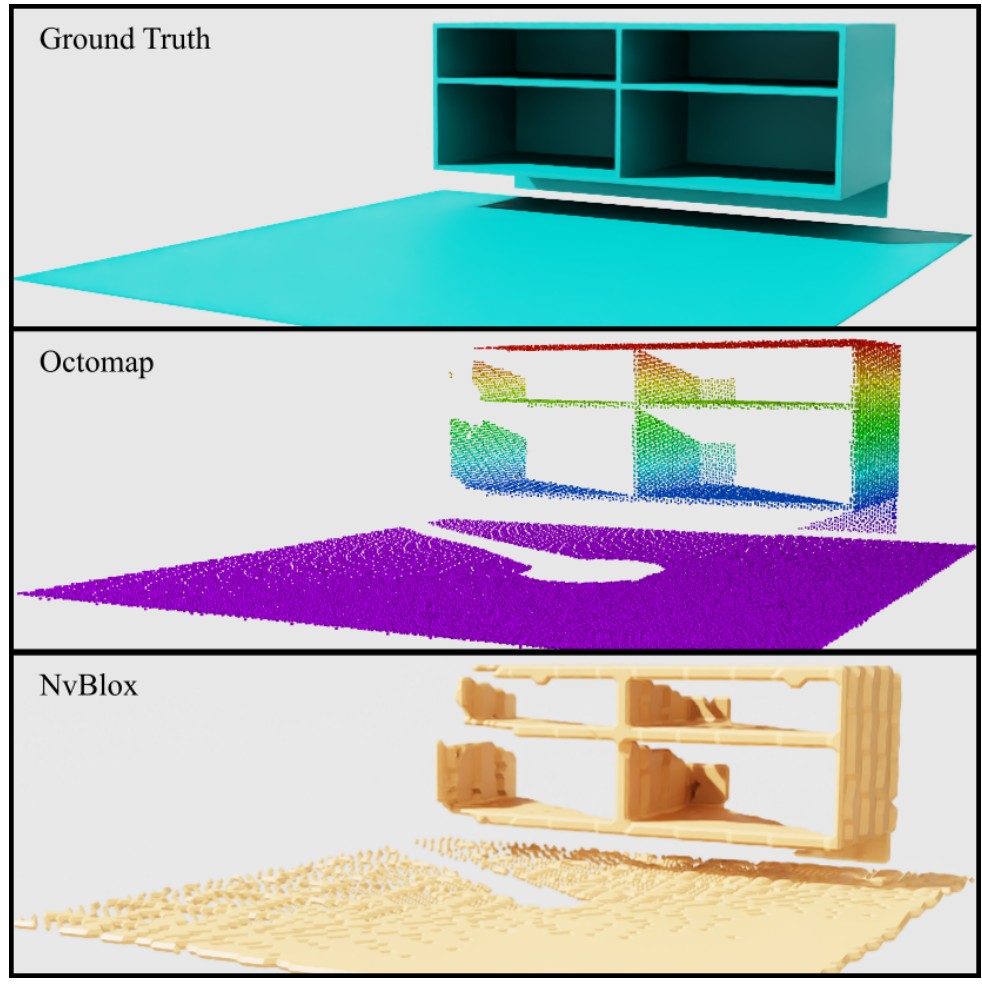

Figure 5: A typical failure case for classical planners is that they do not account for collisions in unobserved regions. In this example, the reconstructions from both Octomap [66] and NvBlox [67] leave large holes due to occlusion. *Avoid Everything* is able to leverage learned priors to produce safe movement without an explicit reconstruction.

still significantly lower than OctoMap without completion, where RSR was $99.62\%$ and SCR was $53.30\%$. Given the performance demonstrated in the original AdaPoinTr publication [68], we suspect that this performance could be significantly improved by retraining the model on a selection of our scenes, but due to resource constraints, we leave this investigation to future work.

## J   Maintaining Reaching Performance After Fine Tuning

**ROPE**   We aimed to determine the efficacy of *ROPE* by varying the ratio of hard negative examples in each fine-tuning batch. We set this parameter $r$ as a constant value for the entire fine-tuning procedure and studied how different values change the performance (see Figure 7). For these experiments, we looked only at the cubby setting and used fully observed point clouds, similar to those used during training. We observed a monotonic decrease in collision rate as $r$ increased. However, we also observed a monotonic increase in the reaching error, *i.e.* the minimum distance from the target after rolling out for 70 time steps. With no fine-tuning, we measured an average reaching error of $0.58$cm and a collision rate of $6.43\%$. At $r = 20\%$, we observe an average reaching error of $0.64$cm with a collision rate of $2.37\%$. At $r = 50\%$, collision rate is below $1\%$, but reaching error averages $1.41$cm. We chose $r = 20\%$ for our other experiments, but the choice of this parameter should be determined by the downstream application and the criticality of collision

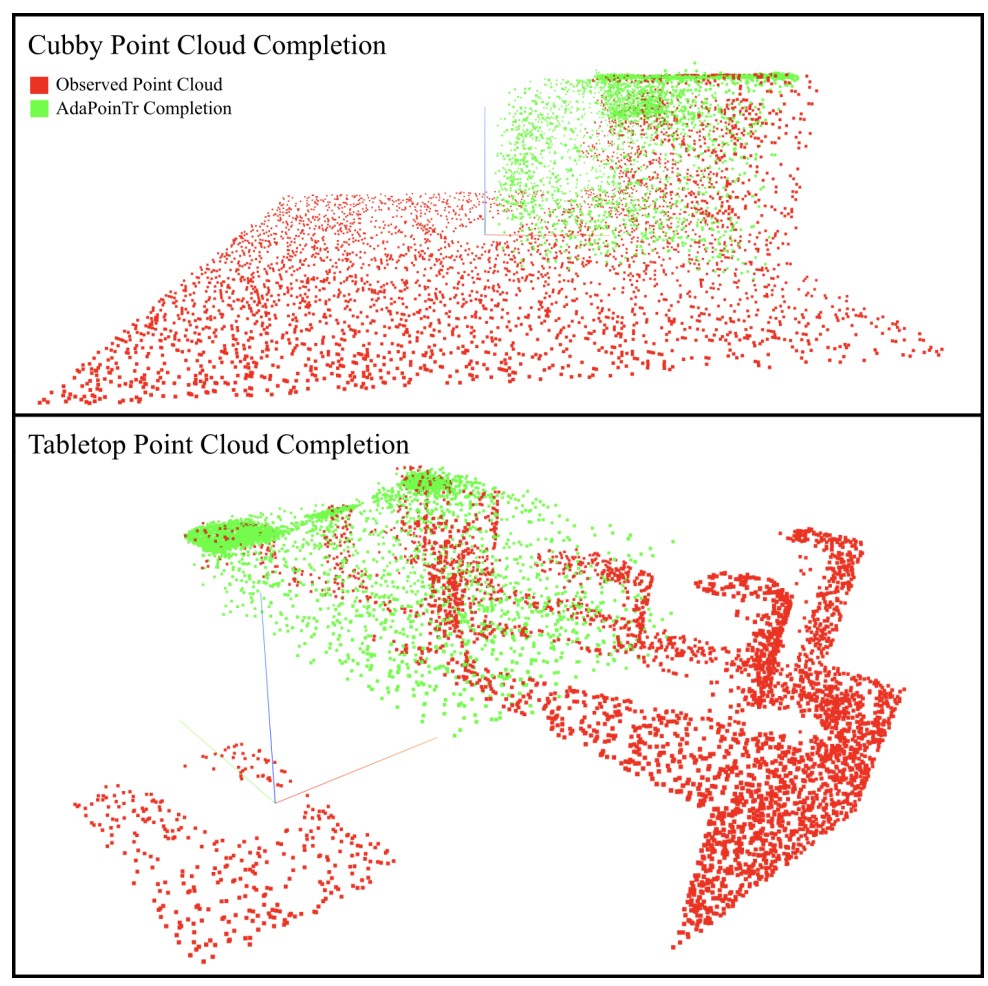

Figure 6: Learned Point cloud completion is a common technique to address unobserved regions of a point cloud. However, when we used the pretrained state-of-the-art point cloud completion network AdaPoinTr [68], we found that it produced highly inaccurate results for our scenes, likely due to distribution shift. In the cubby scene (top), the point cloud completion adds volume to the front of the cubby, making it hard to plan. In the tabletop scene (bottom), the completion misses a large portion of the scene and fails to capture the geometry of the objects.

avoidance. We did not experiment with varying $r$ during fine-tuning as a function of performance, but we hypothesize that setting it as a function of performance would improve results.

**DAgger**    One of the most common techniques for fine-tuning a learned policy is DAgger[18]. DAgger aids in accounting for distribution shift by asking the expert to provide demonstrations at every state the pretrained policy would visit. Likewise, *ROPE* can be seen as a way to account for distribution shift by correcting the policy when it fails. While DAgger is a generally useful tool for imitation learning, it requires making many costly calls to the expert. In our case, each expert demonstration requires 20 seconds of computation time, which adds up quickly if a demonstration is needed at every state visited by the policy. We implemented two versions of DAgger as comparisons and show the performance in Table 3. In the first version, we ran the pretrained *Avoid Everything* through its entire training data, collected the trajectories with collisions, and requested an expert demonstration at every step leading up the collision. We found that this technique can improve performance, reducing the pretrained collision rate of 6.43% in cubby setting to 4.08%, but it is not better than *ROPE*, which reduces the collision rate to 2.37%. We attribute this to the fact that the DAgger corrections use the same expert, which often veers very close (5mm) to obstacles. To verify this, we tested a second version of DAgger that uses a more conservative expert for

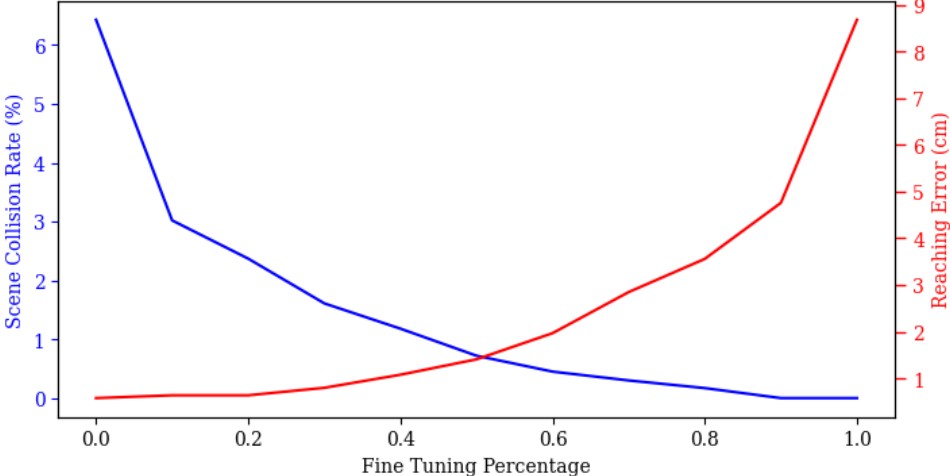

Figure 7: Fine-tuning can be run with different proportions $r$ of hard negative examples. As $r$ increases, the collision rate goes down and target error increases. We attribute this phenomenon to the model overfitting to the hard negatives and forgetting the original behavior cloning objective.

corrections–one with a 2cm collision buffer. We label this more conservative technique *Cons. DAgger* in Table 3. As discussed in Section 4, this expert is more limited in the problems it can solve, *e.g.* not those that either start or end within 2cm of obstacles. However, we found that this version of DAgger significantly improves collision avoidance without negatively impacting reaching performance, dropping collision rate in the cubby setting to 1.28%. We observe a similar drop in the tabletop setting, bringing pretrained collision rate from 11.26% to 2.31%. Running DAgger, however, is very computationally intensive—collecting DAgger demonstrations for the policy's failures on our training dataset required nearly five days on a desktop with an NVIDIA 3090 GPU and an AMD Ryzen Threadripper 3990X 64-Core Processor.

When used alone, *ROPE* outperformed DAgger with the original 5mm expert in both the cubby and tabletop settings. Meanwhile, fine-tuning with *Cons. Dagger* outperforms both. However, we did not find *ROPE* to be to be mutually exclusive of DAgger. With both versions of DAgger, we were able to further improve performance by using *ROPE* as a second fine-tuning step. The best performance came from stacking the conservative DAgger technique with *ROPE*, with success rates of 95.71% and 91.97% in the cubby and tabletop settings respectively.

## K    Real Robot Experiments

We used a dual-computer setup running ROS to control our Franka Emika Panda robot. The control computer, which runs a real-time linux kernel, has Intel(R) Core(TM) i7-4770 CPU with 16 Gigabytes of RAM. The second computer, which runs *Avoid Everything*, has an Intel(R) Core(TM) i9-9900K CPU, 32 Gigabytes of RAM, and an NVIDIA Titan RTX GPU. We use a Kinect V2 for perception, which captures point clouds at approximately 10Hz. We use [89] for eye-on-hand calibration and [90] to remove the robot from the depth cloud; we then re-insert these robot points into the cloud using the deterministic sampling method described in Section F. We are able to run the model at approximately 25Hz on our hardware, which allows for reactive motion. We send each predicted action directly to a lower level joint controller [91].

The model is able to react to moving obstacles in the scene, but due to speed of our camera, it can take up to 140ms—100ms for the camera update, 40ms for the model update—for the robot to react to an obstacle. We expect that this reactivity could be improved with a faster camera, a faster GPU, or both. We used our best performing checkpoint, which was first fine-tuned with the conservative DAgger pipeline and then fine-tuned with *ROPE* (see Section 5.1.4).

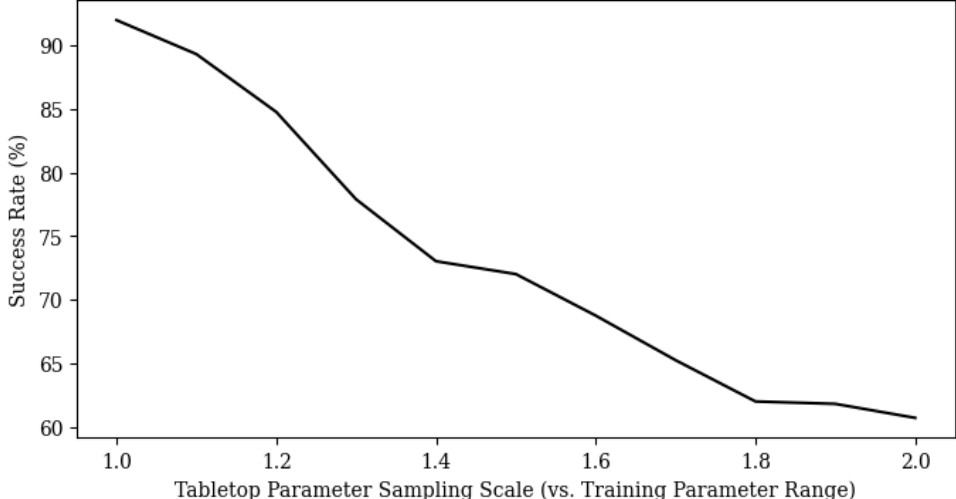

Figure 8: As the distribution of environments expands beyond the training distribution, *Avoid Everything*;s performance degrades. We evaluated this by evaluating *Avoid Everything* on equally sizes evaluation sets with increasingly wide distributions of parameter values. On the far left is performance on a test set drawn from the same distribution used to generate the training data and on the far right is a test set where all parameters are drawn randomly from a range twice as wide as the corresponding range used to generate the training data.

One challenge in our setup is that the gripper of the Franka is nearly symmetric about the axis that points from the wrist to the midpoint of the fingers. Our training data consisted of randomly generated poses, but these poses typically sampled from only half of the rotations about this axis. When we provided an out-of-distribution pose where the $180°$ rotation about this axis would be in distribution, we observed the robot typically tries to exploit the symmetry of the gripper and reach the symmetric in-distribution pose. Depending on the application, these $180°$ rotations may or may not be acceptable. We believe this could be fixed by increasing the variation of target poses in the training set, adding a unique per-point embedding to the gripper points to distinguish orientations, or both.

## L   Generalization

Despite *Avoid Everything*'s strong performance on in-distribution environments, we found that the performance does degrade in environments that lie outside of the training distribution. To understand the rate of decay, we generated ten additional sets of 10,000 environments and planning problems with increasing randomness. As described in Section 4, our environments are generated according to procedural rules using randomly sampled parameter values. For each parameter, we scaled the range from which it could be drawn. As these scaled values go up, the sampled environments are more-and-more out of distribution. We found that while performance does decay, the network is still able to safely solve many out-of-distribution problems (see Figure 8). In environments that are much further out of distribution, however, we observed that our system does not generalize. To test this, we evaluated the model trained on tabletop environments on our test set of tabletop environments and found the model to success in only $8.61\%$ of problems. We hypothesize that co-training on many classes of environments as in [13] would lead to stronger generalization, but due to our computational constraints, we leave this to future work.

