# OpenReview forum: "Avoid Everything: Model-Free Collision Avoidance with Expert-Guided Fine-Tuning"
_robot-learning.org/CoRL/2024/Conference — CoRL 2024_

### Official Review · Reviewer_Ue3E · 2024-07-16
**Review of Avoid Everything: Model-Free Collision Avoidance with Expert-Guided Fine-Tuning**

**Originality:** 4
**Technical Quality:** 4
**Clarity Of Presentation:** 4
**Potential Impact:** 3
**Recommendation:** 3
**Confidence:** 4

**Review:**

Strength:
1. The paper is well-written and easy to follow. The video attachment clearly captures the decent performance of the proposed method.
2. The M$\pi$Former design that learns the delta joint angles from scene observation and target gripper position is logically sound.
3. The end-to-end Data-driven motion planner helps the robot to do real-time planning and re-planning, which has a decent potential contribution to the field.
4. The authors are welcome to include detailed information about the key elements in the proposed method such as network architecture, data generation process, learning platform, and the pseudocode for ROPE.
5. The experiment section is comprehensive and compares the proposed method with both data-driven and classical sampling-based methods.

Weakness:
1. A brief introduction about the scene observation part should be added to the method part to make it more complete.
2. Just like the author mentioned in the limitation section, the end-to-end method cannot guarantee that the resulting path is collision-free. The method can be further enhanced by introducing classic pipelines at the point where the planner fails just like what MPNet [1] does.

[1] A. H. Qureshi, Y. Miao, A. Simeonov and M. C. Yip, "Motion Planning Networks: Bridging the Gap Between Learning-Based and Classical Motion Planners," in IEEE Transactions on Robotics, vol. 37, no. 1, pp. 48-66, Feb. 2021, doi: 10.1109/TRO.2020.3006716.
keywords: {Planning;Robots;Neural networks;Path planning;Search methods;Training data;Probabilistic logic;Deep learning in robotics and automation;learning and adaptive systems;learning from demonstration;motion and path planning},



Minor Comments:
1. Section 4, line 182, the original authors call their method as ABIT* instead of AIT*.

**Quality Of The Limitations Section:**

3

**Questions For Rebuttal:**

The authors are encouraged to address the following questions/concerns:
1. In Section 3.2, line 154, when you do roll-out on the training data, does it mean you apply the pre-trained M$\pi$Former starting from some randomly selected states of the training trajectory for a fixed horizon? This part is not very clear to me.
2. I think one advantage of the proposed data-driven motion planner is that it is very fast and thus suitable for real-time applications. I wonder why the authors chose not to include planning time as one of the metrics.
3. It might be better to include ABIT* in the baselines so that the readers can better validate the training performance.

**Robotics Focus:**

4

**Summary Of Paper:**

This paper proposes a novel end-to-end motion planning approach named avoid everything. The authors design a motion policy transformer that predicts the delta joint angles based on the scene observation and the target gripper pose. They also come up with a ROPE strategy to generate feasible robot motion in complex environments, which boosts training performance. The proposed method beats the baselines in simulation environments in terms of success rate and scene collision rate. The design is also verified in real-world robot manipulators.

**Summary Of Recommendation:**

I rate this paper as weak accept. The paper is well-written and contains enough information/evidence to properly evaluate the proposed method. The end-to-end data-driven motion planner contributes to the real-time reactive motion planning field. The limitations of the proposed method are also well-addressed.

---

### Official Review · Reviewer_xqeu · 2024-07-20
**Avoid Everything: Model-Free Collision Avoidance with Expert-Guided Fine-Tuning**

**Originality:** 2
**Technical Quality:** 4
**Clarity Of Presentation:** 4
**Potential Impact:** 2
**Recommendation:** 2
**Confidence:** 3

**Review:**

Avoid Everything is a very well-written paper that tackles the challenge of developing motion control policies that can avoid collisions in cluttered environments. Importantly, Avoid Everything learns to avoid collisions from visual representations of partially observed scenes.


**Strengths**
1. The authors demonstrate their approach working well in simulation and on hardware.
2. The authors say that a limitation of their approach is that it can only run at 33Hz. This is actually quite fast for robot planning, and thus is a strength of their approach.


**Weaknesses**
* Some relevant works are missing from the related works: learned safety representations (e.g. neural SDFs), reachability analysis, trajectory optimization methods
* The motivation for the network inputs could be better. For example, why are you sampling points from the robot mesh? Why not over approximate the robot mesh with standard collision primitives such as spheres, capsules, cubes, or other polytopes? It's not clear if sampling the robot mesh is leading to an unnecessarily complicated architecture or if sampling is the cause of the collisions that are seen in the results.
* The success rate is fairly high, but the scenarios are also fairly easy. It seems like transformers are an overpowered tool for these tasks and perhaps a more traditional model-based method would work as well.
* Curobo is the only comparison to a model-based method. What about other methods such as CHOMP or TrajOpt. Also, what would happen if you allowed curobo to plan in a receding horizon fashion? Maybe it would be less likely to have collisions? Maybe you can allow curobo to have some fine-tuning?

**Quality Of The Limitations Section:**

3

**Questions For Rebuttal:**

* Can the author include more literature review on provably safe planning? I believe it will better contextualize this work within the broader robot safety community.
* Can the authors justify their approach better? For example, why learn an implicit safety representation? Why not learn an explicit safety representation and plug that representation into a motion planning algorithm?
* Can this method be considered end-to-end if it requires a fine-tuning step?
* How does the fine-tuning step work? The authors say it happens online. Does this mean real-time? Does it happen on the robot?
* Could you provide timings for the IAT* planner? Does it work with point clouds or some other scene representation? Is it fast? If so, why is Avoid Everything needed?
* The authors should discuss how they are performing collision detection. Is it discrete time or continuous time?
* Can the authors explain why it is sufficient to have a loss function that only penalizes collisions? It means that the Avoid Everything can’t actually avoid everything. In other words, the title of the paper is misleading considering the vast amount of work that has been done on safe planning and controls.
* Can the authors explain why they don't compare to a baseline that enforces hard collision avoidance constraints?
* The authors compare to M\piNets, which they state, “which is state of the art for learned end-to-end collision-free motion”. Why not compare to actual state-of-the-art collision-free motion planning methods?

**Robotics Focus:**

4

**Summary Of Paper:**

This paper develops an approach called Avoid Everything for learning transformer-based motion planning policies for serial manipulators in cluttered scenes. Importantly, the authors train a model using visual input (point clouds).

**Summary Of Recommendation:**

I suggest a weak reject of the paper because the claims of Avoid Everything are not adequately demonstrated by the authors.

---

### Official Review · Reviewer_SUwx · 2024-07-21
**Fine-tuning phase alleviates collision avoidance, the positive effects of applying the method on object centric manipulation tasks is not clear**

**Originality:** 3
**Technical Quality:** 3
**Clarity Of Presentation:** 4
**Potential Impact:** 2
**Recommendation:** 3
**Confidence:** 3

**Review:**

Quality

The paper presents high-quality research, characterized by thorough experimentation and detailed analysis. The proposed system, Avoid Everything, is rigorously evaluated through extensive simulations and real-world tests, ensuring the reliability and robustness of the results. The combination of the Motion Policy Transformer (MπFormer) and the ROPE fine-tuning technique is well-implemented, showcasing significant improvements over existing methods.

Clarity

The paper is generally well-written and organized. The methodology is clearly described, with sufficient details provided for both the architecture and the fine-tuning process. The use of figures, tables, and pseudocode enhances understanding, although some complex concepts may benefit from additional explanatory text.

Originality

The originality of this work is notable. The introduction of MπFormer and the innovative ROPE fine-tuning method represent significant advancements in the field of robotic motion planning. The approach of leveraging point clouds for end-to-end learning and the use of expert-guided optimization to refine the model are both novel and impactful. This work stands out by addressing the challenge of collision avoidance in partially observed environments, a persistent issue in robotics.

Significance

The significance of this research lies in its potential to advance the capabilities of robotic systems in real-world applications. By achieving a high success rate in challenging environments, the proposed system offers a practical solution for safe and efficient robotic manipulation. The improvements over previous state-of-the-art methods highlight the impact of this work, making it a valuable contribution to the field of robotics and autonomous systems.

Strengths
Innovative Architecture: The Motion Policy Transformer (MπFormer) introduces a novel way of predicting joint space controls from point clouds, enhancing the model's ability to navigate complex environments.

Effective Fine-Tuning: The ROPE technique effectively reduces collision rates by incorporating hard negatives, demonstrating a practical approach to improving model performance.

Comprehensive Evaluation: The paper includes extensive simulated and real-world experiments, providing robust evidence of the system's effectiveness.

Significant Performance Gains: The system achieves over a 91% success rate, outperforming previous methods and demonstrating its practical applicability.

Weaknesses

Generalization Limitations: While the system performs well in tested scenarios, its ability to generalize to entirely novel environments or tasks not seen during training remains an open question.

**Quality Of The Limitations Section:**

3

**Questions For Rebuttal:**

The paper focus on collision avoidance in goal reaching policy, but does not state clearly how this method can benefit solving manipulation tasks. When a manipulation policy is doing simple tasks such as pick and place using either offline or online policy, how can the proposed method be integrated in such policy and maintain the level of success rate?

**Robotics Focus:**

4

**Summary Of Paper:**

This paper presents a robust end-to-end system for collision-free motion generation in cluttered, partially observed environments. It introduces the Motion Policy Transformer (MπFormer), a transformer-based model trained on over 1,000,000 expert trajectories to predict joint space controls from point cloud inputs. Additionally, the paper proposes Refining on Optimized Policy Experts (ROPE), a fine-tuning technique that corrects collision-prone states through optimization. The system achieves a success rate of over 91% in challenging settings, significantly outperforming previous methods. It demonstrates strong performance and robustness in both simulated and real-world environments, showcasing its potential for practical robotic applications.

**Summary Of Recommendation:**

The paper proposes an end-to-end system that can create safe, collision free motion toward a goal using partially observed point cloud. However, the second fine-tuning phase may impede the performance of a manipulation task, so the positive effects of applying this method in manipulation tasks are not clear

---

### Author Rebuttal · Authors · 2024-08-13

We have attached our rebuttal submission. Our updates are highlighted in red.

[Last Edited August 13, 1:49pm AoE Time]

---

### Decision · Program_Chairs · 2024-09-04

**Decision:**

Accept

**Comment:**

The reviewers appreciate the well-motivated and clearly structured work. However, the reviewers also highlight that the paper would further benefit from additional model-based baselines, evaluations on more complicated scenarios, and the discussion of some additional related work.

## Post Rebuttal
The rebuttal did address most if not all of the major concerns. Although, the very late upload of the rebuttal made it difficult for reviewers to provide further feedback, the additional baselines, evaluation and clarifications certainly improved the manuscript.